# Expression of *INPP5D* Isoforms in Human Brain: Impact of Alzheimer’s Disease Neuropathology and Genetics

**DOI:** 10.3390/genes14030763

**Published:** 2023-03-21

**Authors:** Diana J. Zajac, James Simpson, Eric Zhang, Ishita Parikh, Steven Estus

**Affiliations:** Department of Physiology and Sanders-Brown Center on Aging, University of Kentucky, Lexington, KY 40508, USA

**Keywords:** *INPP5D*, SHIP1, Alzheimer’s disease, next-generation sequencing, allelic expression imbalance, single nucleotide polymorphism, genetics, microglia

## Abstract

The single nucleotide polymorphisms rs35349669 and rs10933431 within Inositol Polyphosphate-5-Phosphatase D (*INPP5D*) are strongly associated with Alzheimer’s Disease risk. To better understand *INPP5D* expression in the brain, we investigated *INPP5D* isoform expression as a function of rs35349669 and rs10933431, as well as Alzheimer’s disease neuropathology, by qPCR and isoform-specific primers. In addition, *INPP5D* allelic expression imbalance was evaluated relative to rs1141328 within exon 1. Expression of *INPP5D* isoforms associated with transcription start sites in exon 1 and intron 14 was increased in individuals with high Alzheimer’s disease neuropathology. In addition, a novel variant with 47bp lacking from exon 12 increased expression in Alzheimer’s Disease brains, accounting for 13% of total *INPP5D* expression, and was found to undergo nonsense-mediated decay. Although inter-individual variation obscured a possible polymorphism effect on *INPP5D* isoform expression as measured by qPCR, rs35349669 was associated with rs1141328 allelic expression imbalance, suggesting that rs35349669 is significantly associated with full-length *INPP5D* isoform expression. In summary, expression of *INPP5D* isoforms with start sites in exon 1 and intron 14 are increased in brains with high Alzheimer’s Disease neuropathology, a novel isoform lacking the phosphatase domain was significantly increased with the disease, and the polymorphism rs35349669 correlates with allele-specific full-length *INPP5D* expression.

## 1. Introduction

Sixty to 80% of Alzheimer’s Disease (AD) risk is linked to genetics [1]. Genome-wide association studies (GWAS) for AD have identified multiple single nucleotide polymorphisms (SNPs) in genes associated with AD risk, a large subset of which are primarily expressed in microglia: *INPP5D*, *TREM2*, *CD33*, *PLCG2*, *CASS4*, *HLA-DRB5-DRB1*, *LILRB2*, *MS4A4A*, *ABI3*, *SCIMP*, and *PTK2B* [2,3,4,5,6] and reviewed in [7] and [8]. In AD, microglia accumulate around amyloid plaques in the brain and phagocytose neuronal debris, taking on a transcriptional signature termed “disease-associated microglia” (DAM) (reviewed in [9,10]). A gene necessary for the microglial transition to the DAM transcriptional phenotype is the Triggering-Receptor Expressed on Myeloid cells-2 (*TREM2*). The AD-associated loss-of-function R47H variant of *TREM2* increases proinflammatory signatures of microglia and hyperactivation of AKT in AD individuals and in a tauopathy mouse model [11]. Higher levels of soluble TREM2 are associated with slower amyloid accumulation in amyloid-positive individuals and slower tau deposition and cognitive decline in amyloid and tau-positive individuals [12]. Soluble TREM2 increases when the receptor is cleaved off the cell surface upon receptor activation, which indicates that increased TREM2 signaling correlates with reduced amyloid and tau deposition in humans. However, conflicting results suggest the need for further study of TREM2 activation at the early and late stages of the disease. When injecting AD-tau into an amyloid-murine model, chronic activation of TREM2 with an activating TREM2 antibody increased amyloid load and tau-phosphorylation, indicating that chronic activation of TREM2 in late-stage AD may exacerbate the disease [13]. Hence, increasing TREM2 signaling in pre or early AD may be a viable therapeutic approach for decreasing AD risk or slowing AD progression. Interestingly, Inositol Polyphosphate-5-Phosphatase D (*INPP5D*) that encodes the SHIP1 protein, which is an inhibitor of the TREM2 signaling pathway, has two AD-associated SNPs identified by GWAS [14,15].

*INPP5D* is a gene that is rising to the forefront of AD-genetics research because one of its AD-associated SNPs, rs35349669, is the most common AD-risk variant, accounting for 3.8% of all genetic risk for AD [16]. *INPP5D* encodes the SHIP1 protein, which has two primary functional domains. The Src-homology 2 (SH2) binding domain is responsible for SHIP1 migration to the cell membrane to colocalize with activated TREM2 and its co-receptor DAP12 (also known as TYROBP) [17]. The SHIP1 phosphatase domain acts to dephosphorylate secondary messengers and thereby inhibit pathway signaling [17]. SHIP1 inhibits signal transduction downstream of immunoreceptor tyrosine-based motif (ITAM)-containing proteins associated with several receptors, including TREM2, FcεR1γ, and B cell antigen receptor (BCR) [17]. The negative regulatory action of SHIP1 on microglial TREM2 signaling is through the phosphoinositide kinase (PI3K) pathway. The PI3K pathway is responsible for the phosphorylation of phosphatidylinositol-4, 5-bisphosphate (PI-3,4-P_2_) to the secondary messenger phosphatidylinositol-3, 4, 5-triphosphate (PIP_3_), which signals downstream to increase microglial proliferation, mobility, and phagocytosis [18,19]. SHIP1 acts to dephosphorylate PIP_3_ to PI-4,5-P_2_, thereby inhibiting PI3K downstream signaling, which is necessary for microglial activation to the DAM transcriptional state [20]. In addition to the negative regulation of PI3K action, SHIP1 also inhibits Syk by competing for its binding to phosphorylated DAP12 [17]. Although the physiological role of SHIP1 in AD is under investigation in murine studies downregulating SHIP1 expression, currently published reports on these mice show inconsistent results [5,21,22].

GWAS identifies variants that are associated with the genetic risk of disease and have identified two SNPs in *INPP5D* associated with AD risk. The minor allele of rs35349669 in intron 10 is associated with increased AD risk (*p* = 4.85 × 10^−9^, minor allele frequency (MAF) = 0.498, Z-score = 5.85). Conversely, the minor allele of rs10933431 in intron 2 is associated with decreased AD risk (nominal *p*-values from Jansen summary statistics (*p* = 8.92 × 10^−10^, MAF = 0.220, Z-score = −6.13)) [14,15]. The two SNPs are in poor linkage disequilibrium (LD) with an R^2^ of 0.18 in the CEU population [23]. Many SNPs are candidates to mediate the rs35349669 association with AD because it is in strong LD with many SNPs, including synonymous SNPs within exons seven and ten, but rs10933431 is not coinherited with other SNPs with an LD greater than 0.82 [23]. Currently, the functional effects of neither AD-associated SNP have been identified in relation to *INPP5D* gene expression or AD pathology.

In this study, we quantify the expression of several previously identified, as well as novel *INPP5D* isoforms in post-mortem brain tissue of aged AD and non-AD individuals. We evaluate the relationship between *INPP5D* isoform expression and AD neuropathology, as well as SNP effects. SNP allelic eQTLs (expression quantitative trait loci) were identified by using next-generation sequencing to quantify unequal allelic expression. To our knowledge, this study is the first to evaluate and compare *INPP5D* isoform expression targeting multiple transcription start sites in human brains and the first to identify a significant rs35349669 allele-specific SNP effect on *INPP5D* expression.

## 2. Materials and Methods

### 2.1. DNA and RNA Extraction from Human Brain Tissue

The RNA, genomic DNA and cDNA samples used for this study have been extensively described previously [24,25,26,27]. Tissue samples from the anterior cingulate cortex of AD and non-AD individuals were provided by the University of Kentucky AD Center Neuropathology Core. The anterior cingulate cortex was used because the area is moderately affected in AD but does not have extensive neuronal death that could skew cell type-specific RNA proportions [28,29]. The National Institute on Aging-Reagan Institute (NIARI) criteria for neuropathological diagnosis of AD based on amyloid and tau deposition were used as a measure of AD neuropathology, where scores of “intermediate likelihood” or below were defined as low AD neuropathology, and scores of “high likelihood” were defined as high AD neuropathology (reviewed in [30]). All samples with high AD neuropathology had a Braak score of VI. The overall dataset included 28 AD samples (12 male and 16 female) and 28 non-AD samples (14 male and 14 female). The age at death for AD individuals was 82 ± 6 years (mean ± SD, *n* = 28), and for non-AD individuals was 82 ± 9 (mean ± SD, *n* = 28). The post-mortem interval (PMI) for AD individuals was 3.4 ± 0.6 h, and for non-AD individuals was 2.8 ± 0.9.

### 2.2. PCR Amplification and Quantitation

The human DNA samples were genotyped for the AD-associated SNPs rs35349669 and rs10933431 and the AEI reporter SNP rs1141328 by using allele-specific TaqMan FAM and VIC dye-labeled genotyping kits (ThermoFisher, Waltham, MA, USA) on a real-time PCR machine as directed by the manufacturer: denaturation at 95 °C for 10 min, and PCR cycling at 95 °C, 15 s; 60 °C, 1 min; 40 cycles. The distribution of AD SNP genotypes for these individuals can be seen in Table 1. For the AEI reporter SNP rs1141328 assay, a subset of 26 heterozygous samples was used and included 11 AD (5 male, 6 female) and 15 non-AD (8 male and 7 female) samples.

The cDNA copy numbers for each sample were determined by quantitative polymerase chain reaction (qPCR) relative to standard curves that were amplified in parallel and were based upon previously purified and quantified PCR products, as described previously [27,31,32,33]. *INPP5D* isoforms were identified using ENSEMBL (version GRCh38:CM000664.2) and are referred to with ENSEMBL nomenclature. Genetic sequences corresponding to *INPP5D* isoforms were used to generate appropriate qPCR primers. To identify *INPP5D* splice isoforms, cDNA from 10 AD and 10 non-AD samples was mixed and used as a template for PCR primers targeting 300–800 base pair (bp) *INPP5D* fragments using Q5 High Fidelity HotStart DNA Polymerase (NEB) as directed by manufacturer: initial denaturing at 98 °C for 30 s, and PCR cycling at 98 °C, 10 s; between 65 and 67 °C (depending on recommended for specific primers), 15 s; 27 cycles. PCR-amplified products were separated on a 10% w/v polyacrylamide gel (Acrylamide: Bis-acrylamide 29:1; Biorad) and detected using SYBR Gold fluorescence. PCR products were excised and sequenced (ACGT Inc., Chicago, IL, USA) to identify *INPP5D* isoforms.

Figure 1 illustrates the isoforms captured by each primer pair. The exon 2 and exon 3 primer pair targets only isoforms containing the SH2 binding domain, which are the full-length isoforms (201 and 204) and the truncated non-protein-coding 205 isoform. The primer pairs corresponding to exon 10 and exon 12 target only full-length protein-coding isoforms that include both the SH2 binding domain and phosphatase domain (201, 204).

The primer pairs corresponding to sequences within exon 11 and exon 12 target protein coding isoforms containing the phosphatase domain (isoforms 201, 202, 204). The intron 14 and exon 16 primer pair targets only the 213 isoform, which is not protein-coding. Primers against exons 15 and 16 target all isoforms containing the phosphatase domain (201, 202, 204, and 213). Transcription start sites (TSS) A, B, and C (TSS-A, TSS-B, TSS-C) are illustrated with yellow letters. TSS-A starts at exon 1 and corresponds to isoforms 201, 204 and 205, TSS-B starts at exon 11 and corresponds to isoform 202, and TSS-C starts at intron 14 and corresponds to the 213 isoform. A full list of the primers and their sequences are presented in the Appendix A.

### 2.3. Allelic Expression Imbalance

The AEI approach was previously described by Parikh et al. [34]. Here, we used rs1141328 as the AEI reporter SNP. This SNP is in the 5′ untranslated region (UTR) and is not coinherited with either AD SNP. For AEI analysis, samples were amplified for 28 cycles with primers corresponding to exon 1 and exon 2 for cDNA or exon 1 and intron 1 for genomic DNA (gDNA). The reactions used Q5 High Fidelity HotStart DNA Polymerase (NEB), and cycling conditions consisted of initial denaturation at 98 °C for 30 s, cycling at 98 °C for 10 s, annealing between 63 and 70 °C (depending on primer) for 15 s, and extension at 72 °C for 30 s, with a final extension of 72 °C for 10 min. A total of 2.5 µL of this reaction was subjected to a subsequent PCR to attach sequencing adaptors. This 25 µL reaction consisted of an initial denaturation of 98 °C for 30 s, 2 cycles of 98 °C for 10 s, between 63 and 70 °C (depending on primer) for 15 s, and extension at 72 °C for 30 s, followed by 8 cycles of 98 °C for 15 s; and 72 °C for 15 s, and a final extension of 72 °C for 10 min. PCR-amplified products were separated on a 10% polyacrylamide gel, visualized with SYBR Gold fluorescence, and the amplicons were excised, eluted [35], and purified with a Monarch PCR and DNA Cleanup kit (NEB). Exon numbering is according to the ENST00000445964.6 transcript on ENSEMBL, and primers are listed in the Appendix A. Following a Bioanalyzer QC, the PCR amplicons underwent Illumina 2 × 250 MiSeq DNA amplicon Next-Generation sequencing (UK Healthcare Genomics Core Laboratory). This process yielded a total of 13,156,256 reads, with about 3.5 million reads identified as our amplicons. Barcodes separated reads into each genomic and cDNA sample. A 16-mer sequence (GCCGGCCC[G/A]GCCGAGG), where brackets denote rs1141328 alleles, was used in an R script to count copies of each allele.

### 2.4. INPP5D Isoform Stability

U937 cells (ATCC) were maintained in RPMI 1640 with HEPES (Invitrogen 42401-018) with 10% *v*/*v* fetal calf serum (characterized, Low LPS), 50 U/mL penicillin and 50 µg/mL streptomycin. Cells were maintained in a humidified 5% carbon dioxide/95% room air atmosphere. A total of 10^5^ cells were plated in 0.9 mL media in a 24-well plate. Cycloheximide (CHX) was dissolved in ethanol at 50 mg/mL and diluted to 5 mg/mL in Hank’s Balanced Salt Solution (HBSS). Cells were treated with CHX (50 μg/mL) or vehicle control for 1, 3, 5, and 8 h in triplicate. Cell suspensions were centrifuged at 300× *g* for 5 min, and RNA was extracted from cell pellets using RNeasy kits according to the manufacturer’s instructions (Life Technologies, Carlsbad, CA USA). RNA was converted to cDNA by using random hexamers and SuperScript IV (Thermo, Waltham, MA USA) and subjected to PCR as described previously for human brain cDNA [27,31,32,33].

### 2.5. Statistics

The copy numbers for each qPCR product were log-transformed to normalize the data. The expression of *INPP5D* isoforms was compared to the expression of Integrin Subunit Alpha M (ITGAM), a known microglial gene. To compare *INPP5D* isoforms against each other, we compared each isoform against the exon 10 to exon 12 qPCR product because this product captures all *INPP5D* isoform expression that includes both the SH2 and the phosphatase domain. Effects of AD pathology and genetics were evaluated by using linear regression analyses (SPSS v28). Effects of SNPs on *INPP5D* expression in the AEI assay were assessed by non-parametric Mann-Whitney U test (SPSS v28). Unless otherwise indicated, all *p* values are nominal *p* values.

## 3. Results

### 3.1. Quantitation of INPP5D Isoforms as a Function of AD Neuropathology and Genetics

Since exons 10 and 12 are present in each of the isoforms that encode full-length SHIP1 (Figure 1), we quantified the expression of the full-length *INPP5D* isoforms using primers corresponding to exons 10 and 12. Since *INPP5D* expression is largely restricted to microglia in the brain [36], *INPP5D* copy numbers were analyzed relative to *ITGAM* expression (Figure 2). We found that full-length *INPP5D* isoforms are significantly upregulated in individuals with high AD neuropathology (adj R^2^ = 0.917, NIARI *p* = 0.008). However, significant correlations between *INPP5D* expression and either AD SNP were not detected (*p* > 0.05).

Next, we asked whether isoform-specific expression of *INPP5D* is associated with either AD neuropathology or the SNPs of interest. We targeted isoforms corresponding to the three TSSs in *INPP5D*; (i) primers against exon 2 and exon 3 target isoforms that begin at TSS-A (ENSEMBL 201, 204, and 205 (Figure 3A–C)), (ii) primers against exon 11 and exon 12 target isoforms that begin at TSS-B or TSS-A (201, 204, and 202 (Figure 3D–F)), and (iii) primers against intron 14 and exon 16 target the isoform that begins at TSS-C (213, Figure 3G–I). For each assay, we found that *INPP5D* isoform expression increased with AD neuropathology. More specifically, the isoforms corresponding to TSS-A captured by exon 2 to exon 3 increased expression in AD (Figure 3A, adj R^2^ = 0.828, NIARI *p* = 6.14 × 10^−^^5^). Isoforms captured by exon 11 to exon 12 also were increased in expression with high AD neuropathology (Figure 3D, adj R^2^ = 0.896, NIARI *p* = 0.005). Lastly, isoforms corresponding to TSS-C captured by intron 15 to exon 16 increased expression in AD (Figure 3G, adj R^2^ = 0.682, NIARI *p* = 1.45 × 10^−4^). However, an SNP effect on expression was not discerned for each of these isoforms (*p* > 0.05).

To compare the expression of *INPP5D* isoforms further, we used full-length *INPP5D* isoform expression to normalize the quantitation data for each TSS (Figure 4). We found that TSS-A (exon 2 to exon 3) and TSS-C (intron 14 to exon 16) isoforms are upregulated in AD compared to full-length isoforms (Figure 4A, adj R^2^ = 0.914, NIARI *p* = 8.56 × 10^−4^; and Figure 4G, adj R^2^ = 0.728, NIARI *p* = 0.002, respectively). Because the primer pair targeting TSS-A captures both FL and SH2-only isoforms, and the primer pair targeting TSS-C captures the truncated phosphatase-only isoforms (Figure 1), these data suggest that enriched expression of partial-length *INPP5D* isoforms 205 and 213 is associated with greater AD neuropathology. The TSS-B isoform (exon 11 to exon 12) shows no difference in expression between high and low AD neuropathology from that of full-length isoforms, indicating that the 202 isoform captured by TSS-B and lacking the SH2 domain is not significantly changed with AD relative to full-length *INPP5D*. No significant SNP effect on *INPP5D* TSS expression was observed (*p* > 0.05).

### 3.2. Identification of Novel INPP5D Isoforms and Quantitation with AD Neuropathology

Since expression of the major *INPP5D* isoforms was not associated with AD genetics, we hypothesized that the SNPs might act via a rare *INPP5D* isoform. One example of a rare *INPP5D* isoform is the skipping of exon 26 (GTEx [37]). Skipping of exon 26 (D26) deletes the carboxyl-terminal 198 amino acids of SHIP1 and alters the codon reading frame such that SHIP1 ends in a novel 17 amino acid carboxyl tail (SEALSELPLSREPRGTA). Since initial work indicated that D26 was quite rare, we generated a qPCR assay for the *INPP5D* isoform that contains exon 26. This effort confirmed that the exon 26 retained isoform was the primary isoform present and, when compared against the ex15-16 isoform, averaged 104.7 ± 19.2 (mean ± SD). Expression of this isoform was not associated with AD or AD genetics (Figure 5).

To comprehensively interrogate whether novel *INPP5D* isoforms are present in the human brain, we used PCR to amplify overlapping segments of *INPP5D* cDNA. These PCR products were then separated by polyacrylamide gel electrophoresis and visualized by SYBR-Gold fluorescence. This effort identified a novel isoform that lacks the initial 47bp of exon 12 (D47) (Figure 6, lanes 4 and 5). Skipping this 47 bp segment results in a frameshift and the introduction of a premature termination codon (PTC). The protein encoded by D47 would consist of the SH2 domain without the phosphatase domain (Figure 1). To quantify D47, we used a forward primer that included the novel junction formed by exon 11 and the 48th bp of exon 12 and a reverse primer in exon 13. We found that D47 expression correlated well with the expression of exon 11 to exon 12 *INPP5D* and represented an average of 13% of *INPP5D* expression (Figure 5D). Interestingly, the expression of D47 relative to total exon 12-expressing *INPP5D* increased significantly with increased AD neuropathology (adj R^2^ = 0.535, *p* = 0.034). However, the proportion of *INPP5D* expressed as D47 was not associated with AD genetics (*p* > 0.05).

### 3.3. Cycloheximide Assay to Quantify Variants Undergoing Nonsense-Mediated Decay

Given the introduction of a PTC in the D47 isoform, we hypothesized that the D47 mRNA might undergo nonsense-mediated decay (NMD). To test this hypothesis, we treated U937 monocytic cells with cycloheximide (CHX), a translation inhibitor that blocks NMD. Cells were treated with CHX or solvent control for 1 h, 3 h, 5 h, and 8 h in triplicate. The amount of D47 relative to total *INPP5D* increased with the duration of CHX treatment, supporting the hypothesis that D47 undergoes NMD (Figure 7). Hence, D47 appears to undergo NMD, and our finding here that D47 represents 13% of *INPP5D* expression may be an underestimate of the transcribed amounts of this isoform.

### 3.4. Allelic Expression Imbalance for SNP-Associated Effects on Gene Expression

We hypothesized that inter-individual variation might obscure our ability to discern a significant SNP effect on *INPP5D* isoform expression. We, therefore, proceeded to use an allelic expression imbalance (AEI) assay to evaluate allele-specific, intra-individual SNP effects on *INPP5D* expression. The rs35349669 SNP is associated with an increased risk of AD, while the rs10933431 SNP is associated with protection against AD. These SNPs are in poor LD with each other (R^2^ of 0.18), the rs35349669 SNP having a minor allele frequency (MAF) of about 50% in the CEU population, and the rs10933431 SNP having a MAF of about 22% in the CEU population. The number of samples in our study corresponding to each genotype is listed in Table 1. The poor LD between rs35349669 and rs10933431 means that we cannot readily predict haplotypes in samples that are heterozygous for both SNPs. Since we cannot discern whether the SNPs are coinherited on the same allele, where they may have opposing effects on expression, or are coinherited on different alleles, where they may have an additive effect on total *INPP5D* expression in an individual, we evaluated rs35349669 effects on allele expression in samples that were homozygous for rs10933431. Similarly, possible effects of rs10933431 on AEI were evaluated in samples that were homozygous for rs35349669.

For this AEI study, we chose to use an SNP in the 5′UTR, rs1141328, as a reporter because of its high minor allele frequency. We produced cDNA and genomic DNA amplicons in samples heterozygous for this reporter SNP and used next-generation sequencing to quantify the number of copies of each allele present. We normalized our counts to that of the genomic DNA control samples (mean ratio rs1141328 = 0.99 ± 0.05) and measured the absolute deviation from the mean. Using the genotypic calls for each AD SNP, we found a significant association of rs35349669 (*p* = 0.017) but not rs10933431 (*p* > 0.05) with *INPP5D* allelic expression (Figure 8). This finding is striking because the reporter SNP in the 5′UTR is well-removed from rs35349669 and SNPs in robust LD with rs35349669 that are located in exons 7 and 10, as well as introns 7–12. Hence, elucidation of the mechanism whereby *INPP5D* expression is affected by these SNP(s) is unclear and will require further experimentation.

## 4. Discussion

The aim of this study was to identify the *INPP5D* isoforms expressed in the human brain and the impact of AD neuropathology and genetics on the expression of these isoforms. As such, this study had several primary findings. First, the comparison of *INPP5D* isoforms relative to *ITGAM* indicated that all *INPP5D* isoforms increase expression in individuals with high AD neuropathology. Second, the comparison of *INPP5D* isoform expression relative to the exon 10–exon 12 FL *INPP5D* isoforms found that increased *INPP5D* expression with AD neuropathology was primarily in association with TSS-A and TSS-C. They may reflect a change in transcription start site access or changes in transcription factors in patients with AD neuropathology. Third, we quantified novel isoforms involving a 47 bp deletion in exon 12 and skipping of exon 26 and found that the former appears increased with AD neuropathology. Since D47 also undergoes NMD, this effort identified a possible confounder in our qPCR analyses. Lastly, an SNP effect on *INPP5D* expression was discernable when assayed by AEI, which detects SNP effects within a person. Specifically, the rs35349669 AD-risk-associated SNP has a significant effect on *INPP5D* unequal allelic expression. Further studies are required to elucidate the function of these isoforms in relation to AD and to determine the direction of effect of the rs35349669 SNP allelic eQTL on *INPP5D* expression.

Prior *INPP5D* studies have determined that *INPP5D* expression is increased in AD [36] but have not examined the expression of different *INPP5D* isoforms. Here, we show a clear association of increased *INPP5D* isoform expression with high AD neuropathology. Intriguingly, we distinguish that TSSs at the beginning of the gene (TSS-A) and corresponding to the 213 isoform (TSS-C) are more upregulated than isoforms corresponding to the TSS at exon 11 (TSS-B). We speculate that AD patients may have an altered transcription factor profile or open chromatin sites on the *INPP5D* gene that influences access to TSSs. Whether the truncated isoform 213 specific to TSS-C has a functional role in AD pathology progression is yet to be investigated. Our findings support those of Tsai et al., who found that *INPP5D* gene expression was upregulated in late-onset AD (LOAD) in the Accelerating Medicines Partnership for AD (AMP-AD) cohort. Tsai et al. also found that *INPP5D* expression, as discerned with a probe targeting the 3′ end, was increased in 5xFAD mice with age and was positively associated with amyloid plaque density [36].

The mechanism underlying the increased expression of *INPP5D* in AD is unclear. The increase is likely not related to the switch to the DAM phenotype because (i) Keren–Shaul et al. report that murine microglial *INPP5D* expression decreases as microglia progress to a DAM transcriptomic profile [38] and (ii) Olah et al. report variable *INPP5D* expression in humans across microglial homeostatic and DAM phenotypes [39]. The increase may be related to the increased inflammation in AD mediated in part by the transcription factor *PU.1*, which has been implicated in AD by genetics and has been shown to bind to *INPP5D* [8,40]. Further studies are necessary to elucidate the mechanisms underlying the increase in *INPP5D* in AD.

Prior to our study, the effect of AD-associated SNPs on *INPP5D* gene expression was not clear. Here, we quantified *INPP5D* isoform expression relative to the AD-risk SNP rs35349669 and to the AD-protective SNP rs10933431 but did not observe a significant SNP effect. We discovered that complex inter-individual variation impeded our ability to discern a relationship between SNP status and *INPP5D* expression by using qPCR. We tested for AEI by using targeted amplicons and next-generation sequencing technology to obtain counts for each allele in our human data. This method allowed us to discern an effect of rs35349669 but not rs10933431 on *INPP5D* expression. We speculate that the rs35349669 SNP acts to increase *INPP5D* expression because the minor allele of this SNP is listed as being associated with significantly increased *INPP5D* expression in whole blood (*p* = 2.2 × 10^−12^) in GTEx [37]. In our attempts to determine the direction of the effect of the SNP on *INPP5D* expression, we amplified cDNA from the reporter SNP to SNPs in exon 7, and exon 10 that are in good LD with rs35349669 SNPs (exon 7: rs1135173, R^2^ = 0.94; rs36127492; exon 10: R^2^ = 0.94, respectively), but unfortunately were not able to produce sufficient clones for haplotype analysis. A more efficient technique that provides long-read haplotyping is needed to determine the direction of effect rs35349669 has on *INPP5D* expression. Recently, He et al. used data pooled from several brain regions to identify the protective rs10933431 SNP as an allele-specific eQTL, although the direction of this effect was not indicated [41]. Overall, both our results and data from He et al. indicate the complexity of AD-associated SNP effects on *INPP5D* expression and emphasize the need for innovative investigative strategies to understand the molecular mechanism(s) whereby SNPs modulate AD risk.

When investigating the novel D47 isoform, we hypothesized that this truncated isoform might undergo NMD because the loss of 47 nucleotides introduces a PTC. Indeed, we found that the D47 isoform increases relative to total *INPP5D* when cells were treated with an inhibitor of NMD. Considering this result, we propose that variable NMD between brain samples differentially affects D47 levels and may confound our ability to discern an association between overall *INPP5D* expression and AD genetics. Future studies will focus on generating D47 protein and testing for stability and function.

The primary limitations of this study are two-fold. First, the statistical power of the SNP analysis could be improved by increasing the number of samples. Second, long-read sequencing technology could be employed to haplotype the cDNA samples to discern the direction of SNP effect on *INPP5D* expression.

Currently, studies investigating the effects of reduced *INPP5D* expression on amyloid neuropathology in murine models are reporting inconsistent results [5,21,22]. Iguchi et al. used an *INPP5D* haplodeficient *TREM2* loss-of-function mouse model and showed that downregulation of *INPP5D* increased microglial association with amyloid beta plaques and partially restored plaque compaction, compared to control animals with normal *INPP5D* expression [21]. Castranio et al. used an inducible *INPP5D* deletion in an APP/PS1 (termed PSAPP) murine model and found that reduced *INPP5D* increased amyloid beta plaque burden [22]. Similar to Iguchi et al., Castranio et al. also demonstrated that *INPP5D* downregulation increased microglial association with amyloid beta plaques. Lin et al. used an *INPP5D* haplodeficient 5xFAD mouse model and showed that *INPP5D* downregulation increased amyloid beta plaque compaction, increased microglial motility to and phagocytosis of amyloid beta plaques, and had a positive effect on cognition [5]. Overall, the studies by Iguchi et al. and Lin et al. but not Castranio et al. indicate protective effects of reduced *INPP5D* expression in AD murine models. Further work is required specifically to clarify.

Given that SHIP1 acts as a negative regulator of the TREM2 pathway, we expect that the benefits of *INPP5D* downregulation on AD progression may directly correspond to the extent that *TREM2* upregulation ameliorates disease progression. *TREM2* is a necessary activator of the DAM transcriptional profile [38], and early in the disease is beneficial by increasing microglial proliferation, association with amyloid plaques, and reduction of amyloid burden in the brain ([42,43], and reviewed in [44]). However, the role of *TREM2* late in the disease is less clear, where TREM2 activation can ameliorate or exacerbate disease depending on the murine model [13,42,45,46]. Therefore, we speculate that *INPP5D* downregulation would be most beneficial in early AD and less so in the late stages of the disease.

Finally, we propose an updated model for SHIP1 modulation of microglial function (Figure 9). The TREM2 signaling pathway is shown here to represent microglial ITAM signaling in general. Activation of the TREM2 receptor, subsequent receptor coupling, and association with the DAP12 complex cause a Syk-mediated signaling cascade through the PI3K pathway and the production of the secondary messengers IP3 and DAG. This downstream signaling leads to MAPK/Akt and calcium signaling that increases microglial phagocytosis, proliferation, and migration [20]. Downregulation of *INPP5D* has been shown to generally increase microglial association with amyloid beta plaques, increase plaque compaction, and in some cases, increase plaque engulfment and phagocytosis [5,21,22]. In this model figure, SHIP1 is depicted as a negative regulator of these PI3K-mediated processes. Hence, the increased expression of *INPP5D* in AD may exacerbate disease pathology by decreasing PI3K/Syk-dependent microglial functions. As such, the increased expression of *INPP5D* in AD may be an inappropriate compensatory response to chronic disease.

## Figures and Tables

**Figure 1 genes-14-00763-f001:**
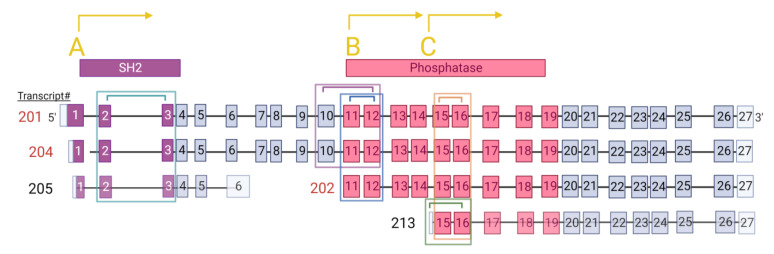
**Illustration of *INPP5D* isoforms captured by different primer pairs.** Primer pairs target full-length isoforms (201 and 204), the SH2 domain isoform (205), and the phosphatase domain isoforms (202 and 203) (isoform names and representation based on ENSEMBL human *INPP5D*). Primer pairs are indicated by the colored brackets, and the corresponding colored rectangles indicate the isoforms captured by those primers: teal = ex2–ex3 (201, 204, and 205); pink = ex10–ex12 (201 and 204); blue = ex11–ex12 (201, 204, 202); orange = ex15–ex16 (201, 204, 202, and 213); green = in14–ex16 (213). Isoforms that are protein-coding are written in red, and non-coding isoforms are faded and written in black. Transcription start sites (TSS)s are marked (**A**–**C**) in yellow and are referred to as TSS-A, TSS-B, and TSS-C, respectively. TSS-A is in exon 1, corresponding to the 201, 204 and 205 isoforms. TSS-B is in exon 11, corresponding to the 202 isoform. TSS-C is in intron 14, corresponding to the 213 isoform. The encoded domains are colored purple for the Src-Homology 2 (SH2) binding domain and pink for the phosphatase functional domain.

**Figure 2 genes-14-00763-f002:**
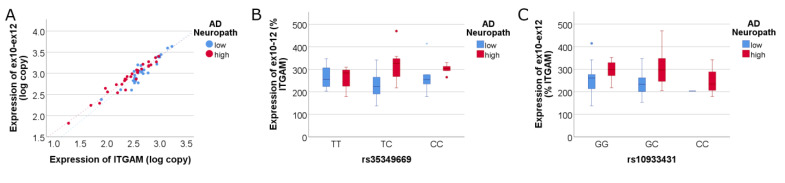
**Expression of full-length *INPP5D* isoforms increases with high AD neuropathology, but a significant SNP effect is not detected**. Full-length isoforms captured by primers targeting exons 10–12 show that *INPP5D* expression is increased in association with high AD pathology (**A**). The linear regression model was highly significant (adj R^2^ = 0.917), where full-length *INPP5D* expression correlated highly significantly with *ITGAM* expression (*p* = 2.14 × 10^−29^; standardized β = 0.99) and was significantly increased with high AD neuropathology (*p* = 0.008; std β = 0.11). Full-length *INPP5D* expression did not significantly correlate with either AD SNP ((**B**,**C**); *p* > 0.05).

**Figure 3 genes-14-00763-f003:**
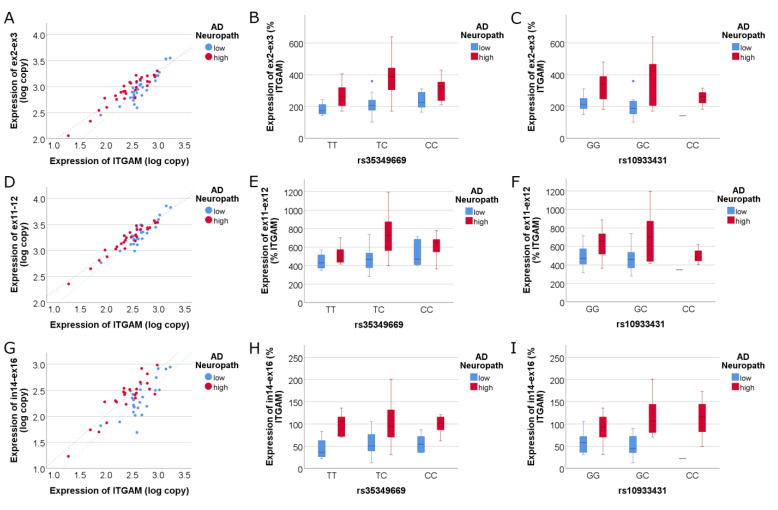
**Isoforms corresponding to all three TSSs increase expression in AD.** (**A**–**C**) depict the isoforms containing the SH2 domain captured by primers targeting exons 2–3. These isoforms increase in expression in association with NIARI neuropathology (**A**). The linear regression model was highly significant (adj R^2^ = 0.828), where ex2–ex3 *INPP5D* expression correlated highly significantly with *ITGAM* expression (*p* = 1.10 × 10^−21^, std β = 0.96) and was significantly increased with high AD neuropathology (*p* = 6.14 × 10^−5^, std β = 0.26). The expression of ex2–ex3 *INPP5D* expression did not correlate with either SNP (*p* > 0.05). Figures (**D**–**F**) depict the isoforms containing the phosphatase domain captured by primers targeting exons 11-12 and show increased expression in AD (**D**). The linear regression model was highly significant (adj R^2^ = 0.896), where ex11–ex12 *INPP5D* expression correlated highly significantly with *ITGAM* expression (*p* = 5.32 × 10^−27^, std β = 0.98) and was significantly increased with high AD neuropathology (*p* = 0.005, std β = 0.14). Figures (**G**–**I**) depict the isoforms transcribed by TSS-C captured by primers targeting intron 14 through exon 16. TSS-C starts at intron 14 and corresponds to the 213 isoform, which lacks the SH2 binding domain and increases expression in AD (**G**). The linear regression model was highly significant (adj R^2^ = 0.682), where in14–ex16 *INPP5D* expression correlated highly significantly with *ITGAM* expression (*p* = 1.54 × 10^−13^, std β = 0.90) and was significantly increased with high AD neuropathology (*p* = 1.45 × 10^−4^, std β = 0.36). The expression of in14–ex16 *INPP5D* expression did not correlate with either SNP (**H**,**I**, *p* > 0.05).

**Figure 4 genes-14-00763-f004:**
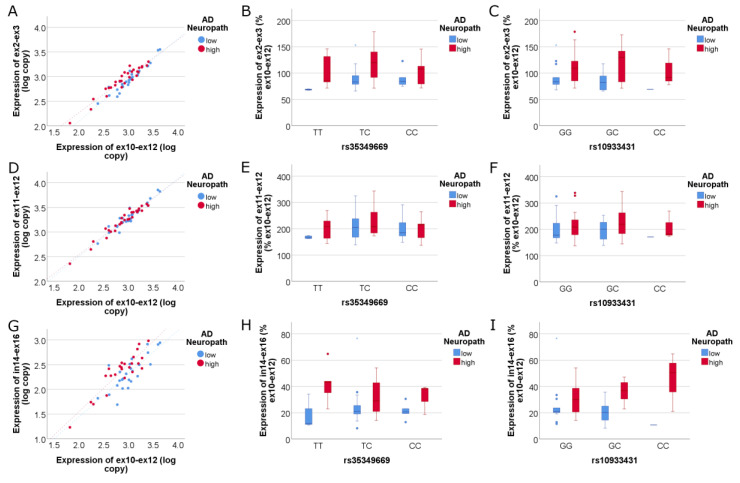
**Expression of isoforms corresponding to all three TSSs as a ratio of full-length (FL) *INPP5D* isoforms.** Graphs (**A**–**C**) depict the isoforms transcribed by TSS-A as a ratio of FL isoforms: Isoforms captured by primers targeting exons 2–3 increase in expression relative to FL isoforms, indicating that the SH2 domain truncated isoform 205 increases in AD (**A**). The linear regression model was highly significant (adj R^2^ = 0.914), where ex2–ex3 *INPP5D* expression correlated highly significantly with ex10–ex12 expression (*p* = 1.72 × 10^−28^, std β = 0.97) and was significantly increased with high AD neuropathology (*p* = 8.56 × 10^−4^; std β = 0.15). Graphs (**D**–**F**) depict the isoforms transcribed by TSS-B relative to FL isoforms: Isoforms captured by primers targeting exons 11–12 have an equivalent expression to FL isoforms, indicating that isoform 202 does not increase expression in AD. The linear regression model was highly significant (adj R^2^ = 0.932), where ex11–ex12 *INPP5D* expression correlated highly significantly with ex10–ex12 expression (*p* = 1.07 × 10^−31^, std β = 0.97) but was not significantly associated with AD neuropathology (*p* > 0.05). Graphs (**G**–**I**) depict the isoforms transcribed by TSS-C relative to FL isoforms, indicating that the 213 isoform captured by intron 14 to exon 16 primers increases expression relative to FL isoforms (**G**). The linear regression model was highly significant (adj R^2^ = 0.728), where in14–ex16 *INPP5D* expression correlated highly significantly with ex10–ex12 expression (*p* = 4.49 × 10^−15^, std β = 0.89) and was significantly increased with AD neuropathology *p* = 0.002). Expression of each of these isoforms was not associated with rs35349669 (**B**,**E**,**H**) or rs10933431 (**C**,**F**,**I**).

**Figure 5 genes-14-00763-f005:**
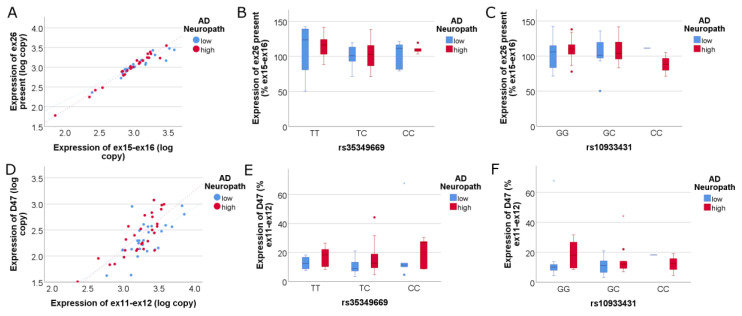
**Expression of the D47 but not the Exon 26 Present *INPP5D* isoform, relative to FL isoforms, show a relationship with AD neuropathology**. Graphs (**A**–**C**) depict exon 26 Present relative to exon 15 to exon 16, which was used as a comparator because the isoforms captured by this assay are those that could capture D26. The assay focuses on Exon 26 Present because preliminary work suggested that the D26 variant may be too rare for accurate quantitation by qPCR. Graphs (**D**–**F**) depict the quantitation of the D47 variant relative to exon 11 to exon 12, which targets all isoforms that could contain D47. Exon 26 present is not significantly associated with AD neuropathology (**A**) nor with either AD-associated SNP (**B**,**C**). The D47 variant normalized to total *INPP5D* expressing D47 (ex11–ex12) has variable expression and yet is significantly associated with AD neuropathology (**D**). The linear regression model was significant (adj R^2^ = 0.535), where D47 *INPP5D* expression correlated highly significantly with ex11–ex12 expression (*p* = 2.32 × 10^−10^, std β = 0.75) and was significantly increased with high AD neuropathology (*p* = 0.034, std β = 0.21). Neither AD SNP is significantly associated with D47 *INPP5D* expression (*p* > 0.05; **E**,**F**).

**Figure 6 genes-14-00763-f006:**
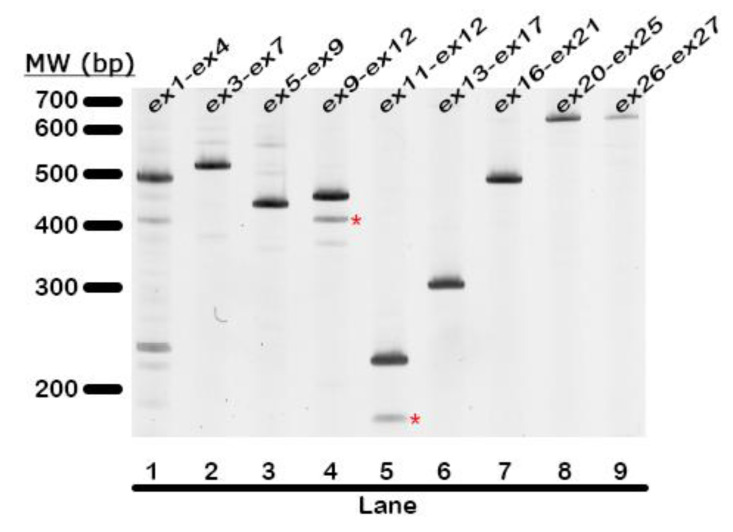
**Search for novel *INPP5D* isoforms identifies D47.** Primer pairs targeting exon 1 through exon 27 were used to PCR-amplify the *INPP5D* sequence throughout the gene to identify novel isoforms present in the human brain. The red asterisks label bands containing a 47bp deletion in exon 12, known as the D47 isoform of *INPP5D*. Lane 1 depicts the full-length 462bp product expected for primers targeting exon 1 to exon 4 with no alterations from prototypic *INPP5D*. Similarly, lane 2 depicts the full-length 547bp product corresponding to primers targeting exon 3 to exon 7. Lane 3 depicts the full-length 444bp product corresponding to primers targeting exon 5 to exon 9. Lane 4 depicts the full-length 458bp product corresponding to primers targeting exon 9 to exon 12, as well as the 411bp product corresponding to D47 (marked with a red asterisk). Lane 5 depicts the full-length 224bp product and D47 (marked with a red asterisk) corresponding to primers targeting exon 11 to exon 12. Lane 6 depicts the full-length 303bp product corresponding to primers targeting exon 13 to exon 17. Lane 7 depicts the full-length 307bp product corresponding to primers targeting exon 16 to exon 21. Lane 8 depicts the full-length 748bp product corresponding to primers targeting exon 20 to exon 25. Lane 9 depicts the full-length 734bp product corresponding to primers targeting exon 26 to the 3’ UTR in exon 27. The full list of primers used can be found in Appendix A.

**Figure 7 genes-14-00763-f007:**
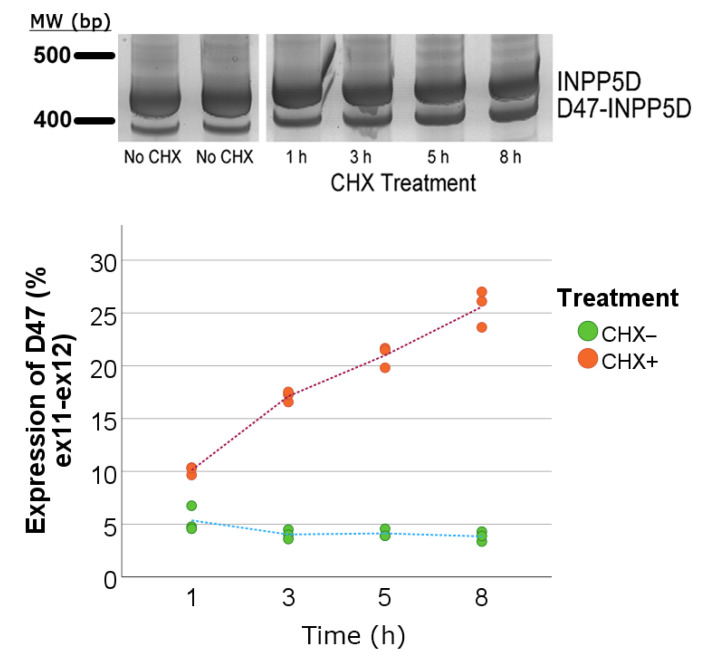
The D47 isoform increases with cycloheximide treatment, as seen by acrylamide gel and qPCR quantitation. The interindividual variability in D47 expression led us to test for NMD of the D47 variant by using a cycloheximide (CHX) assay. U937 cells were treated with either CHX or vehicle control over time. We observed a significant increase in D47 expression as a percent of total ex12-expressing *INPP5D* (ex11–ex12) that positively correlated with CHX treatment over time. Similar results were obtained in an independent replicate of this experiment.

**Figure 8 genes-14-00763-f008:**
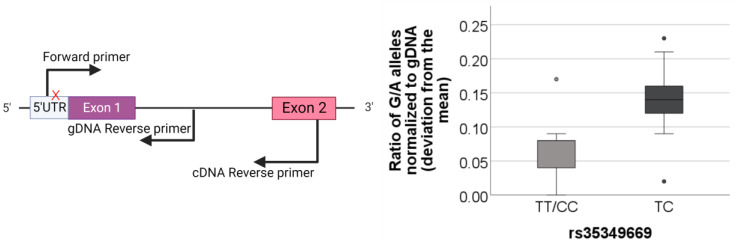
**Allelic expression imbalance quantified by next-generation sequencing of samples heterozygous for reporter SNP.** The left diagram depicts the reporter SNP in the 5’ UTR marked with a red X and the primers used to sequence across cDNA and genomic DNA. The graph depicts quantitative results from next-generation sequencing. The allelic ratio of the G:A rs1141328 alleles, normalized to gDNA, is depicted as a deviation from the mean. To determine the effect of rs35349669 on *INPP5D* expression, only individuals homozygous for the rs10933431 SNP were used (rs35349669: TT/CC *n* = 17, TC *n* = 21). In individuals homozygous for the rs10933431 SNP, the rs35349669 SNP is significantly associated with a difference in *INPP5D* expression, as seen by the difference in rs35349669 homozygous individuals versus heterozygotes. Statistical significance was determined with a non-parametric Mann–Whitney U-test (*p* = 0.017). The rs10933431 SNP was not significantly associated with AEI (*p* > 0.05).

**Figure 9 genes-14-00763-f009:**
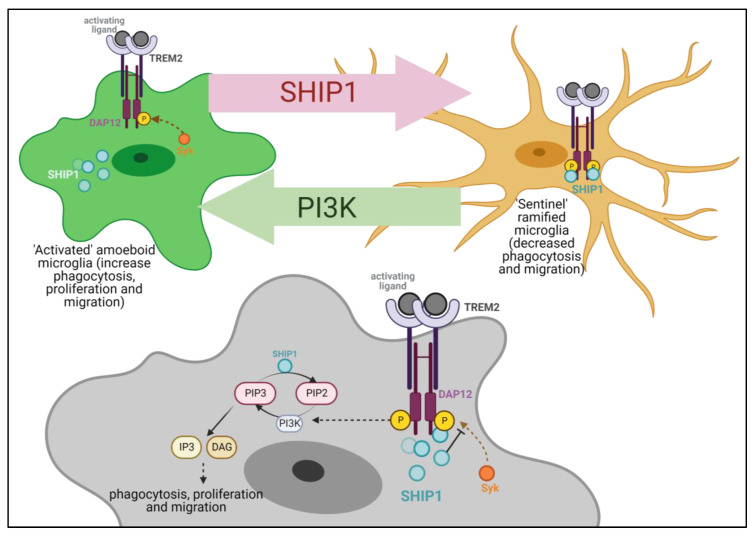
**Molecular model for SHIP1 function in microglial pathway inhibition.** This model depicts the TREM2-DAP12 complex, which is one of the receptor signaling pathways inhibited by SHIP1. Upon TREM2 activation and subsequent phosphorylation of the DAP12 ITAM, SHIP1 moves to the cell membrane. At the cell membrane, SHIP1 inhibits TREM2 signaling by competitively inhibiting Syk binding to p-DAP12 and by inhibiting PI3K activation by dephosphorylating PIP_3_ to PIP_2_.

**Table 1 genes-14-00763-t001:** Numbers of each AD-associated SNP genotypes for in study samples.

rs35349669			
rs10933431	TT	TC	CC
GG	0	2	2
GC	1	15	7
CC	10	19	5

Homozygous minor samples for the AD-associated SNPs are TT for rs35349669 and GG for rs10933431. Samples heterozygous for the AD-associated SNPs are TC for rs35349669 and GC for rs10933431. Samples homozygous major for the AD-associated SNPs are CC for rs35349669 and CC for rs10933431.

## Data Availability

Data are contained within the article or Appendix A.

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
