# Peer review of "Expression of INPP5D Isoforms in Human Brain: Impact of Alzheimer’s Disease Neuropathology and Genetics"

_genes, 2023, doi:10.3390/genes14030763_

Round 1

Reviewer 1 Report

- The title leaves readers bewilder. I highly suggest replacing it with another apt one. 

- Acronyms in the abstract should be avoided.

- The abstract is not informative.

- The innovation of study must be improved.

- Start your discussion by introducing your aim and then discuss the clinical significance of your experiment. Then discuss findings, followed by your study limitations and then conclusion.

- In discussing your findings, compare and contrast them with other studies in the literature and develop arguments and hypotheses for your findings.

- In addition to previous research aligning with yours, please critically discuss those in disagreement and develop arguments and hypothesize. You may also add recommendations.

Author Response

We appreciate the reviewer’s constructive comments.

  1. The title leaves readers bewilder. I highly suggest replacing it with another apt one.

Thank you for the suggestion, we have changed the title.

  1. Acronyms in the abstract should be avoided.

We have removed acronyms.

  1. The abstract is not informative.

We have made edits to the abstract to improve clarity.

  1. The innovation of study must be improved.

We have added text to the introduction to clarify the innovation of the study.

  1. Start your discussion by introducing your aim and then discuss the clinical significance of your experiment. Then discuss findings, followed by your study limitations and then conclusion.

Thank you for the suggestion. We have modified our discussion to be in-line with your suggestion.

  1. In discussing your findings, compare and contrast them with other studies in the and develop arguments and hypotheses for your findings. In addition to previous research aligning with yours, please critically discuss those in disagreement and develop arguments and hypothesize. You may also add recommendations.

This paper is the first to evaluate expression of INPP5D isoforms in the human brain. The only other related paper is from Tsai et al., who looked at overall expression of INPP5D in AD vs non-AD. Since we do discuss the Tsai et al. paper, we have provided a discussion of the most relevant literature that are within the scope of our study.

Reviewer 2 Report

This well written and timely manuscript has several important findings and limitations

PROS

1.       INPP5D isoforms increase expression in brain from patients with AD. This can relate to changes in transcription factors or start site access.

2.       The authors identified novel isoform that appears to be increased with AD neuropathology.

3.       rs35349669 AD-risk associated SNP has a significant effect on INPP5D unequal allelic expression but it hard to say if it is per se functionally related to isoform expression.

4.       Data propose an updated model for SHIP1 modulation of microglial function in AD.

CONS

1.       Limitations of the study should be considered and including consideration disease duration, second neurological and neuropathological hit, replication in other more critical AD related brain areas

2.       Novel isoforms are derived by qRT-PCR only. More robust RNA seq experiments should be considered.

Author Response

We appreciate the reviewer’s constructive comments.

This well written and timely manuscript has several important findings and limitations

PROS

  1. INPP5D isoforms increase expression in brain from patients with AD. This can relate to changes in transcription factors or start site access.
  2. The authors identified novel isoform that appears to be increased with AD neuropathology.
  3. rs35349669 AD-risk associated SNP has a significant effect on INPP5D unequal allelic expression but it hard to say if it is per se functionally related to isoform expression.
  4. Data propose an updated model for SHIP1 modulation of microglial function in AD.

CONS

  1. Limitations of the study should be considered and including consideration disease duration second neurological and neuropathological hit replication in other more critical AD related brain areas.

Thank you for bringing this to our attention. We have added text to address our use of the ACC region. We have also added a paragraph discussing limitations.

  1. Novel isoforms are derived by qRT-PCR only. More robust RNA seq experiments should be considered

We appreciate the suggestion for more robust RNAseq experiments, but in our current experience, the read-depth is insufficient to quantify novel isoforms.

Reviewer 3 Report

Overall, the manuscript is well written although some of the Figures could be made clearer. The findings are significant to the understanding  of INPP5D in AD. The authors propose a testable model/hypothesis in Figure 9.

Line 131/165 should be “10% w/v acrylamide” should think of T% and C% also.

Line 178 “10% v/v fetal calf…”

Line 181 define HBSS

Line 191 define ITGAM

Figure 2 “A linear regression on the ratio between full-length isoform expression to microglial ITGAM expression indicates that the relationship of INPP5D expression to AD pathology is highly significant (adj R2 = 0.917; p = 0.008).” – not clear which one you are referring to. You need to give a better description of what is in the Figure – Figures should be able to be interpreted without resorting to text. Suggest using A, B, C for each subfigure. Suggest placing the linear regression line for “low” and “high “on figure.

Figures 3, 4 & 5 suggest – adding regression line and using A,B,C etc  to designate each graph so legend become clearer

Figure 6 needs a name for the y axis – presume it is (bp)

Figure 10 – y axis is Ratio of G/A alleles yet TT/CC and TC on x axis – keep same for clarity.

Line 315 should be “confounder”

Comment: Remember correlation does not strictly indicate causation and the increased expression of INPP5D could be a sequelae to AD pathology.

Line 352 Which are these data? The findings from the paper or reference 37 – please make clearer.

Author Response

We appreciate the reviewer’s constructive comments.

Overall, the manuscript is well written although some of the Figures could be made clearer. The findings are significant to the understanding  of INPP5D in AD. The authors propose a testable model/hypothesis in Figure 9.

  1.  Line 131/165 should be “10% w/v acrylamide” should think of T% and C% also.
  2. Line 178 “10% v/v fetal calf…”
  3. Line 181 define HBSS
  4. Line 191 define ITGAM
  5. Line 315 should be “confounder”

Thank you for these suggestions. All of the necessary edits to the texts have been made.

  1. Figure 2 “A linear regression on the ratio between full-length isoform expression to microglial ITGAM expression indicates that the relationship of INPP5D expression to AD pathology is highly significant (adj R2 = 0.917; p = 0.008).” – not clear which one you are referring to. You need to give a better description of what is in the Figure – Figures should be able to be interpreted without resorting to text. Suggest using A, B, C for each subfigure. Suggest placing the linear regression line for “low” and “high “on figure.
  2. Figures 3, 4 & 5 suggest – adding regression line and using A,B,C etc  to designate each graph so legend become clearer
  3. Figure 6 needs a name for the y axis – presume it is (bp)
  4. Figure 10 – y axis is Ratio of G/A alleles yet TT/CC and TC on x axis – keep same for clarity.

Thank you for the suggestions. All of the figures have been edited for clarity.

  1. Comment: Remember correlation does not strictly indicate causation and the increased expression of INPP5D could be a sequelae to AD pathology.

This is an interesting point. We have added relevant text to the discussion on Figure 9. We note that since SNPs in INPP5D are associated with AD risk, meaning that SHIP1 plays a role in disease.

  1. Line 352 Which are these data? The findings from the paper or reference 37 – please make clearer.

The text has been edited for clarity.

Round 2

Reviewer 1 Report

In my opinion, Authors haven't justified the innovation of study. Just as title is not conceptual.

Author Response

We thank the reviewer for the feedback, which was “In my opinion, Authors haven't justified the innovation of study. Just as title is not conceptual.”

We apologize for not providing a detailed response to the concern regarding innovation in the first revision.  To be clear, we feel that this study is innovative in three ways. First, this study is the first to identify and quantify expression of multiple INPP5D isoforms in human brain. This effort included a novel isoform that lacks 47 bp from exon 12 and overall established that INPP5D expression is much more complex than previously appreciated.  Second, this study reports that several isoforms, including the novel D47 isoform, are significantly increased with AD neuropathology.  We have interpreted this result as suggesting that AD-associated neuroinflammation upregulates INPP5D expression. Lastly, our allele-specific expression study documents that the AD-associated SNP rs35349669 affects expression of INPP5D isoforms that begin with the prototypic exon 1. This result was striking in part because exon 1 is quite distant from rs35349669 and SNPs in robust linkage disequilibrium with rs35349669. Since rs35349669 is the most common AD genetic risk factor, this finding will be of interest to scientists focused on understanding how AD genetics impacts AD risk. In summary, we submit that this study is innovative with respect to identifying and quantifying multiple INPP5D isoforms, demonstrating selective upregulation of these isoforms with AD, and showing that a prominent AD-associated SNP is associated with altered INPP5D expression. 

We added text to discuss the possible link of INPP5D expression in AD to neuroinflammation, highlighted in yellow:

The mechanism underlying the increased expression of INPP5D in AD is unclear.  The increase is likely not related to the switch to the DAM phenotype because (i) Keren-Shaul et al. reports that murine microglial INPP5D expression decreases as microglia progress to a DAM transcriptomic profile [38] and (ii) Olah et al. reports variable INPP5D expression in humans across microglial homeostatic and DAM phenotypes [39]. The increase may be related to the increased inflammation in AD mediated in part by the transcription factor PU.1, which has been implicated in AD by genetics and has been shown to bind to INPP5D [8,40]. Further studies are necessary to elucidate the mechanisms underlying the increase in INPP5D in AD.

Regarding the concern that the manuscript title is not conceptual, we have changed the title to “Expression of INPP5D isoforms in human brain: impact of Alzheimer’s Disease neuropathology and genetics”. 

In conclusion, we again thank the reviewer for their helpful comments.

Reviewer 2 Report

Improved manuscript

Author Response

Thanks for your help in the review process.